# Differential Expression of Nicotine Acetylcholine Receptors Associates with Human Breast Cancer and Mediates Antitumor Activity of αO-Conotoxin GeXIVA

**DOI:** 10.3390/md18010061

**Published:** 2020-01-17

**Authors:** Zhihua Sun, Manqi Zhangsun, Shuai Dong, Yiqiao Liu, Jiang Qian, Dongting Zhangsun, Sulan Luo

**Affiliations:** Key Laboratory of Tropical Biological Resources of Ministry of Education, Key Laboratory for Marine Drugs of Haikou, School of Life and Pharmaceutical Sciences, Hainan University, Haikou 570228, Hainan, China; zhihuasun918@163.com (Z.S.); zhangsunmanqi@163.com (M.Z.); dongshuai_1024@163.com (S.D.); liuyiqiao.94@foxmail.com (Y.L.); veilness@163.com (J.Q.); zhangsundt@163.com (D.Z.)

**Keywords:** nicotinic acetylcholine receptors (nAChRs), Breast cancer cells, αO-conotoxin GeXIVA, anti-proliferation

## Abstract

Nicotinic acetylcholine receptors (nAChRs) are membrane receptors and play a major role in tumorigenesis and cancer progression. Here, we have investigated the differential expression of nAChR subunits in human breast cancer cell lines and breast epithelial cell lines at mRNA and protein levels and the effects of the αO-conotoxin GeXIVA, antagonist of α9α10 nAChR, on human breast cancer cells. Reverse transcription polymerase chain reaction (PCR) demonstrated that all nAChR subunits, except α6, were expressed in the 20 tested cell lines. Real time quantitative PCR (QRT-PCR) suggested that the mRNA of α5, α7, α9 and β4 nAChR subunits were overexpressed in all the breast cancer cell lines compared with the normal epithelial cell line HS578BST. α9 nAChR was highly expressed in almost all the breast cancer cell lines in comparison to normal cells. The different expression is prominent (*p* < 0.001) as determined by flow cytometry and Western blotting, except for MDA-MB-453 and HCC1395 cell lines. αO-conotoxin GeXIVA that targeted α9α10 nAChR were able to significantly inhibit breast cancer cell proliferation in vitro and merits further investigation as potential agents for targeted therapy.

## 1. Introduction 

Cancer is one of the three principal diseases which lead to human mortality and was responsible for 9.6 million deaths worldwide in 2018 [1]. Globally, nearly 1 in 6 deaths is due to cancer (WHO, 2018) [2]. According to the statistics, breast cancer is the second most common carcinoma in the world following lung cancer and also the highest incidence of cancer in women [1]. In China, Chinese cancer registry annual report in 2015 reported that the incidence of breast cancer is still the first vital type of women cancer with an upward trend [3]. 

Nicotinic acetylcholine receptors (nAChRs), the ligand-activated neurotransmitter receptors in mammals, are divided into muscle and neural nAChRs. The nAChRs are either homomeric or heteromeric pentamers and are comprised of α and non-(alpha) subunits [4,5,6,7,8]. It has been proved that nAChRs are expressed not only in neuronal systems but also in numerous non-neuronal tissues and cells, suggesting that nAChRs may have important roles in other biological processes in addition to synaptic transmission. Further studies showed that nAChRs are related to the development of small-cell and non-small-cell lung carcinomas (NSCLCs), head, neck, gastric, pancreatic, gallbladder, liver, colon, breast, cervical, urinary bladder and kidney cancers [8,9,10,11,12]. Several nAChRs subunits, including α3, α7, α9 and β4, are extensively expressed in various tumor cells and they are involved in regulation of cell proliferation, apoptosis, invasion, migration and angiogenesis [13,14,15,16]. For instance, previous research showed that suppression of α7-nAChRs by snake α-neurotoxins and curare reduced the growth of NSCLC tumor [17,18], Bychkov et al. also confirmed that Ly6/uPAR proteins like Lynx1 retard the cells in G0/1 and G2/M period, inhibit the proliferation and enhance the apoptosis by down-regulating the expression of α7-nAChRs in A549 cells [8]. Furthermore, α-conotoxin AuIB could inhibit small-cell lung carcinomas (SCLC) cell viability via α3/α5/β4-containing nAChRs [12]. As for α9 nAChRs, it is important for promoting cancer cell proliferation, angiogenesis, cancer metastasis and apoptosis suppression during carcinogenesis in response to tumor microenvironments [9,19,20]. So, nAChR antagonists could potentially be used directly or in combination with established chemotherapeutic drugs to treat related cancers.

As competitive antagonists of nAChRs, α-conotoxins (α-Ctxs), have been particularly useful in developing ligands that selectively target one nAChR subtype over another. Some of them have therapeutic potential [21,22,23,24,25]. αO-conotoxin GeXIVA was isolated from *Conus generalis*, which potently blocks α9α10 nAChR [26]. In vivo, it also displayed potent alleviation for neuropathic pain in one rat model [27,28]. Recently, our research also showed that GeXIVA contributed to the inhibition of cervical cancer cell proliferation [29]. αO-conotoxin GeXIVA have the potential for anticancer treatment both by themselves and in combination with other antitumor drugs.

The present study aimed to detect the differential expression of nAChRs in human breast cancer and human normal breast epithelial cells and the nAChR subunits that may be potential targets for breast cancer, to further evaluates the effects of αO-conotoxin GeXIVA on cellular proliferation.

## 2. Results

### 2.1. The LBD cDNA Cloning of Each nAChR Subunit in Various Human Breast Cell Lines

Previous studies demonstrated that nAChRs as ligand-gated ion channels were regulated by binding with acetylcholine (ACh) or nicotinic agonists, such as nicotine and nicotine-containing products [9]. In addition, specific antagonists can block the corresponding different nAChR subtypes at ligand binding domain (LBD) of the receptor. LBD is the important extracellular area for ligands to interact with receptors [30,31]. The LBD cDNAs of each nAChR subunit in 17 cell lines of various human breast cancer and three normal (nonmalignant) human breast cell lines (MCF-10A, MCF-12A and HS578BST) were amplified by reverse-transcription PCR respectively (Figure 1). Each individual agarose gel electropherogram of each nAChR subunit was showed in Appendix A. 

The LBD cDNAs for α3, α4, α5, α7, α9, α10, β2, β3 and β4 neuronal-type nAChR subunits were amplified with expected size respectively in all the 17 breast cancer cell lines and the three human normal mammary gland epithelial cell lines (Figure 1). It means the mRNA of all these nAChR subunits are transcribed constitutionally in both breast cancer cells and normal cells. However, 20 cell lines did not produce a specific amplification fragment for α6 nAChR subunit’s LBD cDNA with the expected size fragments. Therefore, α6 nAChR subunit mRNA displayed in specific cell lines without constitutional expression in all the tested breast cells.

### 2.2. Quantification and Comparison of Different nAChR Subunits mRNA in Different Human Breast Cancer and Normal Cell Lines

As described above, all the tested cell lines were found to express neuronal nAChR subunits of α3, α4, α5, α7, α9, α10, β2, β3 and β4. Moreover, previous studies showed that nAChRs play a crucial role in the course of human cancer development [11,12,32]. In order to quantify and compare the mRNA expression amounts of each nAChR subunit among human breast cancer cell lines and the normal (nonmalignant) human breast cell lines, real-time quantitative fluorescent PCR were used to detect the mRNA expression level of α3, α4, α5, α7, α9, α10, β2, β3 and β4 nAChR subunits respectively (Figure 2). It is obvious that the mRNAs of α5, α7, α9 and β4 nAChR subunits were overexpressed in all the breast cancer cells, which were much higher than that in the normal cell lines HS578BST (Figure 2C,D,E,I). Other subunits, such as α3, α4, α10, β2 and β3 nAChR had similar mRNA expression level (Figure 2A,B,F–H), which expressed higher in part of breast cancer cells than normal cells.

For α5 nAChR subunit (Figure 2C), the highest mRNA expression amount occurred in ZR-75-30 cell line, which had ~400-fold more than that in normal breast cell line HS578BST. The second highest was HCC1395 cells, which had ~230-fold of α5 nAChR subunit mRNA more than that in HS578BST cell line. Then expression of MCF-7 cell line had ~110-fold more than that in HS578BST. There were seven breast cancer cell lines of MDA-MB-361, BT483, SK-BR-3, BT20, AU565, MDA-MB-231 and MDA-MB-157 with 20~60-fold of α5 subunit mRNA higher than that in the HS578BST. The breast cancer cell lines, MDA-MB-453, BT549, HCC1806 and HCC1937 showed 10~20-fold of α5 mRNA higher than that in normal cell line HS578BST. The least α5 mRNA amounts of two breast cancer cell lines, HS578T and Bcap-37 cells had~3-fold more than that in the normal cell line HS578BST, which were statistically significantly different from the normal control. 

Compared with normal cell line of HS578BST, α7 nAChR subunit had the highest expression level of mRNA in HCC1806 cancer cell line with ~600-fold higher than in the normal cell line (Figure 2D). The second highest expression of α7 subunit mRNA was AU565 cancer cells, which had ~120-fold higher expression than the normal cell line. The third highest expression for α7 subunit mRNA was SK-BR-3 cancer cell line, which showed ~52-fold higher expression than the normal cell line. The expression of α7 subunit mRNA of cancer cell lines MDA-MB-157 and MCF-7 showed ~22- fold higher than the normal cell line. The α7 subunit mRNA expression amount of five cancer cell lines that is, BT549, HCC1395, BT20, ZR-75-30 and HCC1937 had more than ~15-fold higher expression than the normal cell line. For MDA-MB-453, MDA-MB-361, Bcap-37, BT483 and MDA-MB-361 cancer cell lines, α7 subunit mRNA expression displayed about 2~10-fold higher than its expression in normal cell line (Figure 2D). 

For α9 nAChR subunit (Figure 2E), the highest mRNA expression amount occurred in MDA-MB-231, which had ~530-fold more than that in the normal breast cell line HS578BST. The second highest was BT549 cells, which had ~300-fold of α9 nAChR subunit mRNA more than that in the HS578BST. Bcap-37 had ~180-fold of α9 subunit mRNA higher than the normal cell line. ZR-75-30 showed ~147-fold higher than the normal cell line. The two breast cancer cell lines, HCC1395 and HCC1806 showed ~38-fold of α9 mRNA higher than that of the normal cell line. There were eight breast cancer cell lines, MDA-MB-453, MDA-MB-361, HS578T, BT483, BT20, MCF-7, AU565 and MDA-MB-157 showed 10~20-fold of α9 mRNA higher than its expression in the normal cell line. For breast cancer cell lines SK-BR-3 and HCC1937, α9 subunit mRNA expression displayed ~4-fold higher levels than normal cell line (Figure 2E).

The most significant upregulation for β4 nAChRs mRNA level was observed in the breast cancer cell line AU565, which had ~1500-fold more than the normal cell line HS578BST. There were three breast cancer cell lines, SK-BR-3, HCC1806 and MDA-MB-231 with 120~300-fold of β4 subunit mRNA higher than that of the HS578BST. Six breast cancer cell lines, MDA-MB-361, Bcap37, HCC1395, ZR-75-30, MCF-7 and HCC1937 showed 25~60-fold of β4 mRNA higher than that of the HS578BST. There was 2~10-fold increase in the amount of β4 nAChRs mRNA in MDA-MB-453, BT549, HS578T, BT483, BT20 and MDA-MB-157 cell lines compared to normal cell line HS578BST (Figure 2I).

Compared with human normal mammary gland epithelial cell line HS578BST, there were 14 breast cancer cell lines expressing 2~20-fold higher levels of α3 nAChR subunit, including MDA-MB-453, BT549, HS578T, Bcap-37, HCC1395, BT483, SK-BR-3, BT20, ZR-75-30, HCC1806, AU565, HCC1937, MDA-MB-231 and MDA-MB-157. MDA-MB-361 and MCF-7 did not show significant changes in α3 nAChR expression from cancer cells compared to normal epithelial cells (Figure 2A). 

For α4 nAChR subunit (Figure 2B), the highest mRNA expression amount occurred in BT20 cells, which had expression of ~145-fold higher than normal breast cell line HS578BST. BT483 and HCC1806 showed expression of α4 mRNA 45~50-fold higher than HS578BST. There were ten breast cancer cell lines, including MDA-MB-453, H578T, Bcap37, HCC1395, SK-BR-3, ZR-75-30, MCF-7, AU565, MDA-MB-231 and MDA-MB-157 with expression of α4 subunit mRNA 2~25-fold higher than normal cell line. BT549, MDA-MB-361 and HCC1937 did not have significant changes shown in α4 nAChR expression from cancer cells compared to normal epithelial cells.

The mRNA expression level of α10 nAChR subunit in five breast cancer cell lines was similar with the normal cell line respectively, which included MDA-MB-453, BT549, SK-BR-3, ZR-75-30 and HCC1937 (Figure 2F). The other 11 cancer cell lines showed expression of α10 mRNA 2~20-fold higher than the normal cell line, which included MDA-MB-361, HS578T, Bcap37, HCC1395, BT483, BT20, HCC1806, MCF-7, AU565, MDA-MB-231and MDA-MB-157 (Figure 2F).

Compared to normal cell line HS578BST, there were nine breast cancer cell lines including MDA-MB-453, Bcap37, HCC1395, BT483, BT20, ZR-75-30, HCC1806, MDA-MB-231 and MDA-MB-157 with expression of β2 nAChR subunit mRNA 4~25-fold higher than HS578BST (Figure 2G). There was no difference in β2 nAChRs subunit mRNA expression between the normal cell line and the seven breast cancer cell lines, that is, BT549, MDA-MB-361, HS578T, SK-BR-3, MCF-7, AU565 and HCC1937 (Figure 2G).

For β3 nAChR subunit mRNA expression (Figure 2H), there were fourteen breast cancer cell lines with 2~20-fold higher than the normal cell line, which were MDA-MB-453, BT549, HS578T, Bcap37, HCC1395, BT483, SK-BR-3, BT20, ZR-75-30, HCC1806, AU565, HCC1937, MDA-MB-231 and MDA-MB-157. Only MDA-MB-361 and MCF-7 cancer cells had no significant change shown in β3 nAChRs expression compared to HS578BST cells.

It is shown from quantitative PCR results that only four nAChR subunits that is, α5, α7, α9 and β4 nAChRs are highly expressed in all tested human breast cancer cell lines compared to normal mammary gland epithelial fibrocystic cells (Table 1).

### 2.3. The Functional nAChRs Expression on the Surface of Breast Cancer Cells and Mammary Gland Epithelial Cells

To confirm the protein expression of nAChRs on the cell membrane, samples of the different cell lines were incubated with fluorescently labelled antibody (FITC-conjugated Goat anti-rabbit Ig (G+L)) and analyzed by flow cytometry (FCM). The specificity of these antibodies was confirmed by Western blotting in some cell lines, such as MCF-7, MAD-MB-157, MDA-MB-231, BT20, SK-BR-3, BT483, HCC1395, MDA-MB-157 α9 KO, HEK-293T, Hela, A549 (Appendix A). Take MDA-MB-157 for example, flow cytometry measurements showed a significant increase in α3, α4, α5, α6, α7, α9, α10, β2, β3 and β4 nAChRs on the surface of breast cancer cells MDA-MB-157 (Figure 3). The difference in mean fluorescence intensity (MFI) for α3 nAChRs between experimental group (MFI 4819) and control group (MFI 115). Likewise, the MFI of α4, α5, α6, α7, α9, α10, β2, β3, β4 nAChRs is significantly higher in MDA-MB-157 cells and the value were MFI 5077, 1060, 1049, 3704, 3653, 4652, 754, 1578, 4852, respectively (Figure 3A). For MDA-MB-157 cells, all investigated functional nAChRs were expressed on the cell surface but the levels varied between nAChR subunits. α3, α4, α7, α9, α10 and β4 nAChRs indicated 32~43-fold difference and α5, α6, β2 and β3 nAChRs were 6~13-fold difference, all the differences had significance (Figure 3B). In addition, protein expression of nAChRs was detected on the all tested cell lines and the results manifested that the nAChRs subunits expressed in all tested cells at protein level (Appendix A). Then, differential expression of each nAChRs was analyzed in different cancer cell lines compared to normal human mammary gland epithelial cells HS578BST (Table 2, Appendix A). It is worth noting that α9 nAChRs were highly expressed in almost all the breast cancer cell lines, when compared to normal cells, difference was prominent (*p* < 0.001) except MDA-MB-453 and HCC1395 cell lines (Table 2).

### 2.4. Protein Expression of α9-Containing Subunit nAChRs in Different Human Breast Cancer Cell Lines

In our study, the mRNA and protein expression differences for the α9 nAChR subunit in different human cancer cell lines have shown in the above results. Thus, further research is warranted to determine the expression differences of α9-containing nAChR protein in these cell lines by Western blot (Figure 4). Using a specific antibody for the α9 nAChR subunit, the total protein content from each cell line was assessed. Expression of α9 nAChR protein was seen in all types of the breast cancer cell lines and normal epithelial cells. Comparing with the normal cells HS578BST, the expression levels of α9 nAChRs protein were significantly higher in most breast cancer cell lines except for MDA-MB-453, HCC1395 and AU565 cancer cell lines (Figure 4). Among them, ten cancer cell lines, including BT20, BT549, BT474, HS578T, Bcap37, MDA-MB-361, HCC1806, HCC1937, ZR-75-30 and MCF-7 with much higher protein expression level than control which were extremely significantly different from HS578BST (*p* < 0.01). There were three cancer cell lines, BT483, MDA-MB-231 and MDA-MB-157 cell lines that were statistically significant *p* < 0.05. 

### 2.5. αO-Conotoxin GeXIVA Affects Breast Cancer Cell Proliferation through the Inhibition of α9-nAChR-Mediated Signals

Previous experiments have established that the over-expression of α9-nAChR at mRNA and protein level in almost all breast cancer compared to normal epithelial cells HS578BST (Figure 2E, Figure 4 and Figure 5) and αO-conotoxin GeXIVA[1,2] was reported that it is a highly potent and selective antagonist of the α9α10 nAChRs subtype [26,28]. α-conotoxin TxID, a selective antagonist of α3β4 nAChRs, served as control. The peptides were obtained by chemical synthesis and two-step oxidation in our laboratory [26,28].

The effects of different treatments on cells proliferation in breast cancer cells were determined by CCK-8 assay. In breast cancer cells, GeXIVA significantly inhibited the proliferation of cancers (Table 3). Analysis of the data showed that the activity of the GeXIVA was concentration-dependent, on breast cancer cell lines, with EC _50_ values ranging from 35 to 130 μM. However, TxID had no effect on breast cancer cell line BT20. Co-treatment of BT20 cells with GeXIVA and TxID resulted in no significant difference of these cells to GeXIVA. Compared with normal cells treated with the same drug at the same concentration, the inhibition of cancer cells at any concentration of GeXIVA used was significantly higher than that of normal cells. 

## 3. Discussion

Recent gene expression profiling studies have categorized breast cancer into four groups including luminal subtypes A and B, HER2+/ER-, Basal-like [32,33] (Table 4). Among them, luminal subtypes A and B were characterized by higher expression of ER; others were characterized by a lack of expression of ER. There are 20 breast cancer cell lines used in this study, which contain nearly all types of known breast cancer cells so far.

The expression of nAChR subunits has been demonstrated in many non-neuronal cells, including human breast epithelial cells and human breast cancer cells [19,33,34]. Previous studies have identified the α9 nAChR over-expression in breast cancer relative to normal tissue [19]. The α7 nAChRs were also found to be overexpressed in cancer stem cells and breast cancer cells, which activated the signaling pathways downstream and made it to be tractable as potential therapeutic targets for breast cancer [35,36]. Chia-Hwa Lee confirmed the expression of α5, α9, α10 nAChR subtypes in normal breast epithelial cells MCF-10A and malignant MCF-7 breast cells, also presented the importance of α9 nAChRs for nicotine induced transformation of normal mammary gland epithelial cells [19,33]. In addition, Mina Kalantari-Dehaghi revealed the expression of genes encoding the neuronal nAChR subunits in MCF-10A and MCF-7 cells, such as α3, α5, α7, α9, α10, β2 nAChR subunits, which can comprise the (α3/α6)β2 ± α5 and α9 ± α10 ACh-gated ion channels and lead to its involvement in the development of cancer [34]. Interestingly, there is a controversy about the expression of α3, α7, β2, β4 subunits in the above two cell lines. For instance, Mina Kalantari-Dehaghi [34] viewed α3, α7 and β2 nAChR subunits exist in MCF-10A cells, contrary to Chia-Hwa Lee [19]. But, the primer sequences used by them are almost identical. Their results were not persuasive. We analyzed their primers and found that the specificity of the two primers is not good by National Center for Biotechnology Information (NCBI) primer-blast, which could lead to appearance of both expected and unexpected fragment. Many breast cancer cell lines have not been investigated yet. To address these questions, we redesign the primers for different nAChR subunits to assay each subunit gene expression level in the 20 cell lines of Table 5. So, this is the first time it is examined each nAChR subunit gene expression (mRNA) in so many (up to 17) breast cancer cell lines and compared with the normal breast cell lines systematically (Table 3). Here it is noteworthy that MCF-10A and MCF-12A cell lines belong to breast epithelial cells but they were derived from woman with fibrocystic disease [37], maybe fibrocystic disease relates to the expression of nAChRs. In this study, the breast epithelial cell line HS578BST was used as control. 

Twenty cell lines were used in this study, there into three cell lines (SK-BR-3, AU565, MDA-MB-453) belonged to HER2 over-expression subtype. Three cell lines (MCF-7, Bcap-37, ZR-75-30) belonged to Luminal A subtype. Two cell lines (BT-474 and MDA-MB-361) fell within subtype of Luminal B. Twelve cell lines (BT-483, BT-549, BT-20, MDA-MB-231, MDA-MB-157, HCC1395, HCC1937, HCC1806, HS578T, MCF-10A, MCF-12A, HS578BST) fell within the Basal-like subtype. We designed the ten neuronal nAChR subunits gene specific primers and demonstrated for the first time that normal human mammary glands epithelial cells and mammary cancer cells express the α3, α4, α5, α6, α7, α9, α10, β2, β3, β4 neuronal nAChR subunits and examined the expression level of each subunit. All the existing neuronal nAChR subunits mRNA were analyzed by RT-PCR, which presented in almost all the cell lines except for α6 nAChR subunit (Figure 1). But, the protein of α6 nAChR was expressed in all cells detected by FCM (Appendix A). It is suggested that there was difference in genome and protein expression level of α6 nAChRs in breast cancer cells. In fact, the details of the relation between mRNA and its encoded protein are not fully clear. The possible reasons for the different expression of the nAChR subunits at two levels include but are not limited to, (1) mRNA stability. In some cases, mRNA degrades rapidly even if it is transcribed, which affects protein expression. (2) A series of powerful and precise regulatory stages, including transcription, post-transcription and translation were involved in Eukaryotic gene expression. Better yet, it is shown from quantitative PCR results that only four nAChR subunits that is, α5, α7, α9 and β4 are highly expressed in all the tested human breast cancer cell lines comparing to normal mammary gland epithelial fibrocystic cells HS578BST (Table 1). Especially α9 nAChR was confirmed by Western blot and FCM (Table 2, Figure 4). The results of this study confirmed the importance of α9-nAChR for breast cancer cells. 

In previous studies, α5 is first considered as an auxiliary subunit that forms functional ion channel of nAChRs only when it is co-expressed with both other α-subunits and β-subunits to a doublet of α3 and either β2 or β4 modifies the pharmacological and biophysical properties of nAChRs and increases Ca^2+^ permeability [9,38]. Then again, recently, researches revealed that a sudden effect of α5-containing nAChRs on nicotine intake was linked to lung cancer cell viability, metastasis and invasiveness [14,39,40]. α5 nAChRs are expressed in all the tested 16 breast cancer cell lines according to our results, which suggested the importance of α5 nAChRs during generation of breast tumors (Table 1). Bychkov et al. and Lyukmanova et al. confirmed that the expression of functional α7-nAChRs on the cancer cells membrane by confocal microscopy [8,11], suggesting that the homomeric channels that had been composed of α7 subunits are a valuable therapeutic target for cancers. The main argument was that nicotine activated α7 nAChRs can cause Ca^2+^ influx into cancer cells and trigger membrane depolarization, which activates voltage-gated Ca^2+^ channels and subsequently activates the MAPK pathway, which may result in inhibition of apoptosis [41,42]. Except for α7 nAChRs, the homopentameric structure of α9 nAChRs has been confirmed to play a major role in breast cancer and other cancer cells [8,11,19,34]. Over-expression and activation of the α9-containing nAChRs during tumorigenesis was discovered in human breast epithelial cells. Moreover, α9 nAChRs were discovered to be expressed higher in high percentage of advanced-stage (stages 3 and 4) breast cancer tissues than in early stages (stages 1 and 2), which occurred more often in active smoker than in passive smoker (secondhand smoke, involuntary smoking and environmental tobacco smoke) [19]. In this study, the mRNA expression level of α9 nAChRs subunit was much higher in all the tested 16 breast cancer cell lines than the normal cell line (Table 1). Besides, the α9 nAChRs protein on the breast cancer cells surface was more than that on the normal cells with significant difference (*p* < 0.001) except MDA-MB-453 and HCC1395 cell lines (Figure 4, Table 2). Then, a stable MDA-MB-157 breast cancer cell line is established in which α9-nAChR was silenced and we found that the suppression ability of αO-conotoxin GeXIVA is decreased, it is crucial to further identify the role of α9 nAChRs in mediating the antitumor activity of αO-Conotoxin GeXIVA.

As antagonists to nAChRs, neurotoxins are normally used to distinguish between neuronal nAChRs receptor subunit combinations due to their ability to selectively interact with receptor subtypes [43,44,45,46]. A recent research focusing on the antagonists that target nAChRs in the treatment of cancer has provided significant insights into their mechanisms of action. Among these nAChRs, the α7-nAChRs are known to be overexpressed in the SCLC of smokers. In vitro experiments have suggested that cancer cells growth can be inhibited using snake neurotoxins (α-neurotoxins) or snail conotoxins (α-conotoxins). Particularly, it has been found that the presence of α7-nAChR inhibitors, such as methyllylcaconitine and α-bungarotoxin, could reverse the pro-angiogenic effects of nicotine and inhibit cancer cell growth. There is moderate evidence suggestive of a critical effect of CHRNA3, CHRNA5 and CHRNB4 expression on SCLC cell viability. α-Conotoxin AuIB is derived from the venom of cone snails and blocks α3β4 nAChRs. Treatment with α-conotoxin AuIB led to decreased viability of DMS-53 cells [12,47]. α-Conotoxin TxID and its more selective analogues [S9A]TxID and [S9K]TxID is newly-found α3β4 nAChRs antagonist with the potency about 60-fold higher than AuIB [48,49,50]. Due to their high specificity and potency on α3β4 nAChRs, TxID, [S9A] TxID and [S9K] TxID from our lab possess great potential as anticancer agent against α3β4 nAChRs overexpressing SCLC. Intriguingly, Qian J et al. indicated that the α3 and β4 nAChRs were upregulated and an antagonist TxID, could inhibit lung cancer cell A549 and NCI-H1299 lines growth [51]. Recently, a series of conotoxins targeting α9α10 nAChRs were discovered and isolated, such as α-conotoxins, Vc1.1, RgIA, It14a and GeXIVA [26,52]. In vivo, α-conotoxin Vc1.1 and RgIA, as well as αO-conotoxin GeXIVA displayed potent alleviation of neuropathic pain in rat model [26,28,53,54]. With regard to GeXIVA, Luo SL et al. investigated the mechanism of blockade of α9α10 nAChRs by GeXIVA[1,2] in Xenopus oocytes and analyzed theinteraction between GeXIVA and the α9α10 nAChR subtype by molecular modeling [26]. Zhangsun DT et al. further demonstrated that aO-Conotoxin GeXIVA disulfide bond isomers retain potency and selectivity for the human a9a10 subtype [55]. In human cervical cancer cell CaSki and SiHa lines, Liu YQ et al. showed that α9-nAChRs were overexpressed compared with that in normal cells [29]. And GeXIVA revealed obvious inhibition of proliferation and its inhibitory effect on normal cells was also significantly less potent than that on cancer cells [29]. It was also shown that α9-nAChRs can be inhibited by some natural compounds. For instance, only low doses of garcinol (1 uM) from the edible fruit Garciniaindica had inhibited nicotine induced breast cancer cell proliferation through the downregulation of α9-nAChR and cyclin D3 expression [35]. Luteolin and quercetin also could inhibit the ability of proliferation by downregulation of the α9-nAChRs expression on cell surface in human breast cancer cells [56]. Tea polyphenol (-)-epigallocatechin-3-gallate have been found to inhibit nicotine-and estrogen-induced α9-nicotinic acetylcholine receptor upregulation in human breast cancer cells and to delay the development of breast cancer cells in vivo [57]. In human cervical cancer cell line CaSki, GeXIVA revealed obvious inhibition of proliferation and its inhibitory effect on normal cells was also significantly less potent than that on cancer cells [29]. In this study, breast cells can be treated with a specific blocker GeXIVA corresponding to α9 nAChRs to further determine the role and significance of these highly expressed α9 nAChRs in breast cancer. Although it also inhibited the normal cells, the effect was significantly less than that on cancer cells. Given these findings, α9α10 nAChRs may be therapeutic targets for breast cancer. Furthermore, α-conotoxins from marine cone snails that inhibit α9α10 nAChRs may provide new clues for breast cancer targeted therapy.

## 4. Materials and Methods

### 4.1. Cell Culture 

The 17 human breast cancer cell lines (Table 4) were purchased from the Kunming Institute of Zoology (Yunnan, China) and American Type Culture Collection (Manassas, VA, USA), which included MDA-MB-157, MDA-MB-231, MDA-MB-361, MDA-MB-453, HCC1395, HCC1937, HCC1806, BT20, BT474, BT483, BT549, HS578T, MCF-7, Bcap-37, SK-BR-3, AU565 and ZR-75-30, the human normal mammary gland epithelial cell lines MCF-10A, MCF-12A and HS578BST. MCF-10A cells were maintained in HuMEC Basal Serum Free Medium supplemented with 10 µL/mL HuMEC Supplement, 50 µg/mL Bovine Pituitary Extract, 200 µL/mL Trypsin/EDTA (Life Technologies, Rockville, MD, USA). MCF-7, SK-BR-3, HS578T, ZR-75-30, BT474, BT483 and HS578BST cells were maintained in Dulbecco’s modified eagle medium (DMEM). MDA-MB-231, MDA-MB-453, MDA-MB-157, MDA-MB-361 were maintained in Leibovitz’s L-15 Medium. HCC1395, HCC1937, HCC1806, AU565, Bcap-37, BT20, BT-549 were maintained in RPMI 1640. MCF-12A was maintained in EMEM. To make the complete growth medium, all the cell were maintained in base medium, supplemented with fetal bovine serum to a final concentration of 10%, 100 U/mL penicillin and 100 mg/mL streptomycin at 37 °C in an atmosphere of 5% CO_2_.

### 4.2. RNA Isolation and RT-PCR Amplification of Ligand Binding Domain (LBD) of Different Human nAChR Subunits

Total RNA of each cell line was isolated using Trizol (Invitrogen, Carlsbad, CA, USA) according to the manufacturer’s protocol. The concentration and purity of RNA were determined by NanoDrop 2000 spectrophotometry (Thermo Scientific, Waltham, MA, USA). Equal amounts of RNA (1 μg) from each sample were reverse-transcribed into first-strand cDNA with High Capacity cDNA Reverse Transcription Kit (Applied Biosystems, Carlsbad, CA, USA), then the first-strand cDNA subsequently amplified by RT-PCR using specific primers. Based on the reported mRNA sequence of ligand binding domain of human (homo sapiens) nAChR subunits, the specific primers for each subunit were designed (Table 5), which were synthesized by Sangon Biotech Co., Ltd. (Shanghai, China). The target fragment was amplified by polymerase chain reaction (PCR) using the cDNA of each cell line as template.

### 4.3. Quantitative Real-Time PCR

Quantitative real-time PCR (qRT-PCR) was performed using Analytikjena qTOWER3G Real-Time PCR system to amplify various nAChR subunits in each cell line, including α3, α4, α5, α7, α9, α10, β2, β3 and β4 nAChR subunits. Each qPCR reaction mix contained 5 µL of SYBR Green I Master (Roche, Indianapolis, USA), 2 µM of each primer, 1 µL cDNA and DNase and RNase-free H_2_O with total volume of 20 µL. The cycling conditions were 95 °C for 10 min (initial denaturation and polymerase activation) followed by 40 cycles of denaturation at 95 °C for 15 s, annealing at 58 °C for 40 s and extension at 60 °C for 60 s. To correct minor variations in mRNA extraction and reverse transcription, the gene expression values were normalized using the housekeeping gene glyceraldehyde-3-phosphate dehydrogenase (GAPDH). Relative quantification of the mRNA level was computed by the comparative Ct (2^−∆∆Ct^) method [58]. The data were analyzed with a sequence detector software (Analytikjena qPCR software 3.2) and nAChR subunits expression levels in breast cancer cells were compared with those in human normal mammary gland epithelial cell line HS578BST. All tests were performed in triplicate.

### 4.4. Protein Extraction and Western Blot Analysis

The cultured cells were washed with cold phosphate buffer saline (PBS) for three times and harvested using a cell lysis buffer containing protease inhibitors PMSF (Solarbio Life Sciences, Beijing, China). Equal amounts of protein from control and treated cell lysates were separated using a 12.5% sodium dodecyl sulfate polyacrylamide gel electrophoresis (SDS-PAGE) gel under reducing conditions and transferred onto polyvinylidene fluoride (PVDF) membranes (Solarbio Life Sciences, Beijing, China) that were subsequently probed with primary antibodies (AChRα9, sc-293282, Santa Cruz, CA, USA). In all Western blot experiments, membranes were additionally probed with an antibody for GAPDH (sc-47724, Santa Cruz, CA, USA) to ensure equal loading of protein among samples. Horseradish peroxidase-conjugated secondary antibodies (m-IgGκ BP-HRP, sc-516102, Santa Cruz, CA, USA) were used with enhanced chemoluminescence reagent (Biosharp, Guangzhou, China) to visualize the protein bands. Images of the films are captured using the Alpha FluorChem E (ProteinSimple, San Jose, CA, USA).

### 4.5. Flow Cytometry Analysis of Functional nAChRs Expression on the Cell Surface

A guava easyCyteTM flow cytometer with InCyte software (EMD Millipore, Hayward, CA, USA) was used to validate the presence of the nAChRs. The cultured cells were harvested and washed with PBS for twice, fixed the cells with 4% PFA for 20 min, washed the cells twice by centrifugation at 1000 rpm for 5 min each time. Suspend the cells in 0.1% Triton X-100 in 1 × PBS buffer for 10 min and washed the cells twice with PBS by centrifugation at 1300 rpm for 5 min each time. The cells were blocked with 3% BSA in 1 × PBS for 30 min and then the cells were washed twice with PBS by centrifugation at 1300 rpm. Add primary antibody at an appropriate dilution and incubate for 2 h at room temperature, washed and incubated with secondary antibody (FITC-conjugated Goat anti-rabbit Ig (G + L)) for 1 h at room temperature. Finally, the cells were washed twice and re-suspended in 300 μL 1 × PBS and the expression of nAChRs was analyzed by flow cytometry. The changes of mean fluorescence intensities of each antibody were calculated and it represent the protein expression level of nAChRs in each cell line. The primary antibodies used were as follows—CHRNA3 Rabbit Polychonal antibody (10333-1-AP, proteintech, Chicago, IL, USA), anti-Nicotinic Acetylcholine Receptor α4 antibody (ab124832, Abcam, Cambridge, UK), anti-CHRNA5 antibody (A02359-2, Boster, Wuhan, China), CHRNA6 Rabbit Polychonal antibody (11388-1-AP, proteintech, Chicago, IL, USA), anti-CHRNA7 antibody (21379-1-AP, proteintech, Chicago, IL, USA), CHRNA9 Rabbit Polychonal antibody (26025-1-AP, proteintech, Chicago, IL, USA), anti-CHRNA10 antibody (ab234767, Abcam, Cambridge, UK), anti-nicotinic acetylcholine receptor β2 antibody (ab55980, Abcam, Cambridge, UK), anti-CHRNB3 antibody (orb338493, biorbyt, Wuhan, China) and CHRNB4 antibody (22192-1-AP, proteintech, Chicago, IL, USA). 

### 4.6. Cell Viability 

The effect of αO-conotoxin GeXIVA on all tested breast cancer cell lines and normal cells HS578BST proliferation were examined by 2-(2-methoxy-4-nitrophenyl)-3-(4-nitrophenyl)-5-(2,4-disulfo-phenyl)-2*H*-tetrazolium, monosodium salt (WST-8) assay kit (CCK-8, Solarbio, Beijing, China). Exponentially growth cells were seeded onto 96-well plates (1 × 10 ^4^ cells per well) and allowed to grow overnight. The cells were treated with five different concentration of αO-conotoxin GeXIVA (180 μM, 90 μM, 45 μM, 22.5 μM and 11.25 μM, respectively). After 24 h, WST-8 was added into each well for 4 h before the measurement according to the manufacturer’s instruction. The absorbance 450 nm was measured by a microplate reader (Molecular Devices, SpectraMax M2, San Jose, CA, USA).

### 4.7. Statistical Analysis

Three to five independent repeats were conducted for all experiments. Error bars represent these repeats. Statistical comparisons between groups were performed using ANOVA (Prism GraphPad Software and SPSS 17.0). A Student’s *t*-test was used and a *p*-value < 0.05 was considered significant. Analysis was performed with the SPSS software package (Version 17.0).

## 5. Conclusions

It is well known that the nAChRs are involved in processes of oncogenesis and inflammation. All the compounds mentioned above provided molecular evidence for the possible chemotherapeutic ability of them for smoking-mediated tumorigenesis through nAChRs pathway. Specifically, α-conotoxins as a common tool in the studies of nAChRs, which blocked various subtypes of nAChRs selectively and potently. In this article, expression of neuronal nAChR subunits in each cell line was delineated through different methods, which suggested α-conotoxins targeting various nAChR subtypes may display potent and additional therapeutic effects on related cancers. This study, the high expression of α9-nAChR in cancer cells was correlated with the pathogenesis. Our study has confirmed αO-conotoxin GeXIVA significantly inhibit growth of cancer cells compared with normal cells HS578BST in vitro. Due to its high potency, GeXIVA may possess great potential as anticancer agent for α9α10 nAChRs overexpressed breast cancer cell lines.

## Figures and Tables

**Figure 1 marinedrugs-18-00061-f001:**
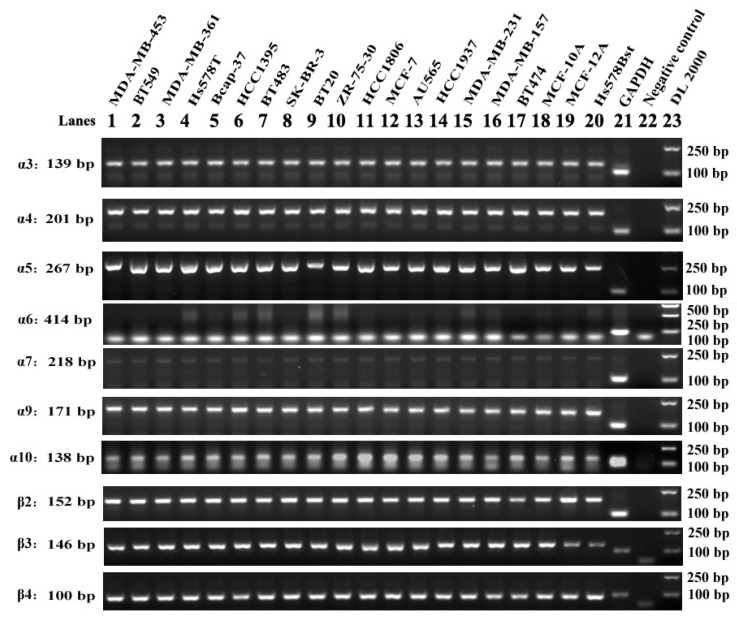
Reverse-transcription polymerase chain reaction (PCR) amplification of neural nAChR sequences from human breast cancer cells and normal (nonmalignant) breast cells RNA. Agarose gel electrophoresis of PCR products amplified from cultured cells RNA samples reverse-transcribed to cDNA. One microgram of purified total RNA extracted from cultured cells were reverse-transcribed to cDNA and amplified for 30 cycles with ten pairs of nAChR subtype-specific primers. The size of each amplification product was determined using the DL2000 DNA Marker ladder standard loaded in lane 23. The bands were consistent with the expected sizes, 139 bp, 201 bp, 267 bp, 218 bp, 171 bp, 152 bp, 146 bp, 100 bp and 103 bp for the α3, α4, α5, α7, α9, α10, β2, β3, β4 neural nAChR subunits (lanes 1~20) and GAPDH (lane 21) respectively, the lane 22 was negative control. Each PCR experiment included one positive control of GAPDH and one negative control of water as template. The negative control could check if the PCR mixture has contamination of other impurity DNA. The positive control of GAPDH gene with a fragment size of 103 bp verified that Reverse-transcription PCR (RT-PCR) was working well, and the synthesized cDNA reverse transcribed from mRNA was completed.

**Figure 2 marinedrugs-18-00061-f002:**
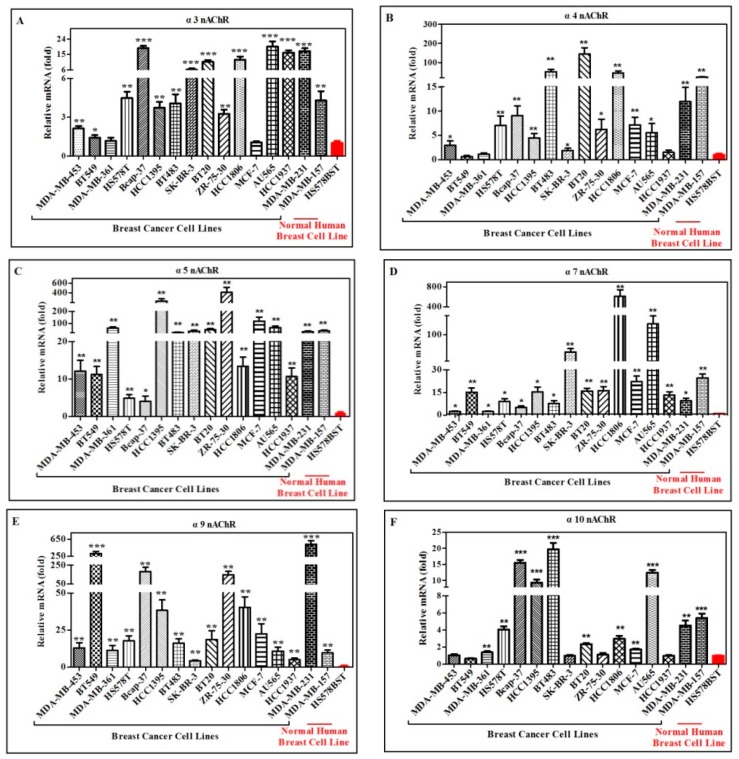
Relative mRNA expression amount of different nAChR subunits in different human breast cancer cell lines and breast epithelial cell line HS578BST (normal control) detected by real-time PCR. Purified total RNA of each cell line with 1 μg was reverse-transcribed to cDNA using poly A primer. Expression level of each nAChR subunit mRNA was examined by real-time PCR using specific primers for each subunit in all the tested cell lines. (**A**–**I**) represent the expression of α3, α4, α5, α7, α9, α10, β2, β3 and β4 nAChR, respectively. * *p* < 0.05, ** *p* < 0.01, *** *p* < 0.001, breast cancer cells vs. normal cells HS578BST.

**Figure 3 marinedrugs-18-00061-f003:**
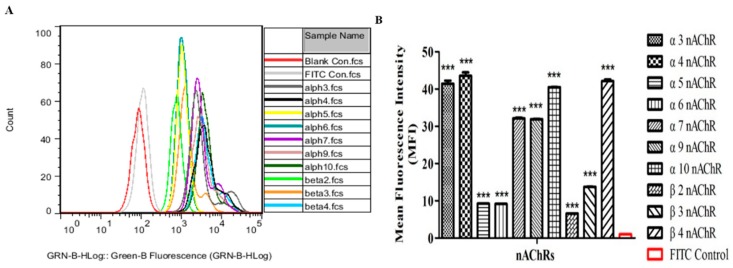
Flow cytometry analysis of nAChRs staining intensity under antibody treatment. (**A**) Histograms of cell distribution according to green fluorescence intensity for cell incubated with anti-nAChRs antibodies and FITC-conjugated Goat anti-rabbit Ig (G + L) are shown. (**B**) Flow cytometry analysis of functional α3, α4, α5, α6, α7, α9, α10, β2, β3 and β4 nAChRs expression on the surface of MDA-MB-157 cells. The mean staining intensities of the cells by fluorescently labelled antibody (FITC-conjugated Goat anti-rabbit Ig (G + L)) are shown. Data are presented as mean fluorescence intensity (MFI) ± SEM, *n* = 3, *** *p* < 0.01, significantly different from the FITC positive control. Statistical analysis was performed with One-way ANOVA test.

**Figure 4 marinedrugs-18-00061-f004:**
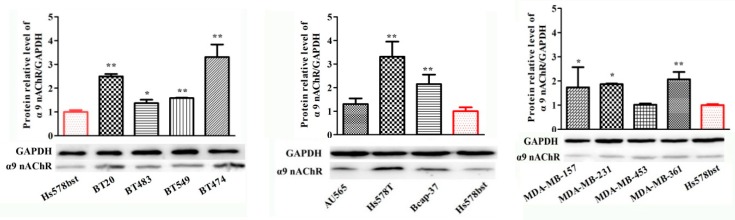
Quantitative analysis of α9-containing nAChRs by Western blot in various cancer cell lines and breast epithelial cell lines. GAPDH was the control in all experiments. The proteins were detected with the corresponding antibody and an anti-mouse IgG secondary antibody conjugated to horseradish peroxidase (HRP). Each bar is the mean ± SD, *n* = 3. * *p* < 0.05, ** *p* < 0.01, breast cancer cell lines vs. breast epithelial cell lines HS578BST (normal cell control).

**Figure 5 marinedrugs-18-00061-f005:**
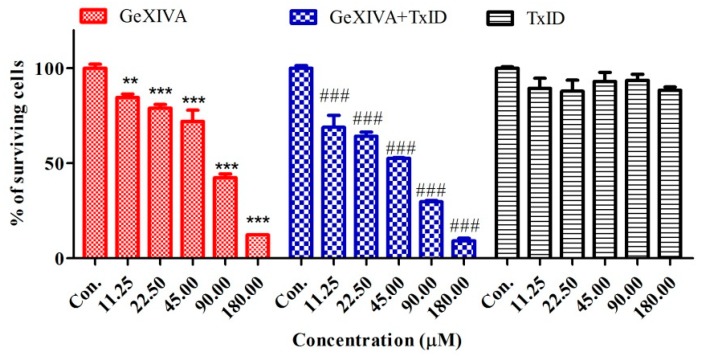
Effects of α-conotoxin on the viability of breast cancer cells BT20. The cells were treated with various concentrations of α-conotoxin for 24 h. Then, the cell viability was determined by the CCK-8 assay. Values were expressed as mean ± SD of three independent assays. Statistical analysis was performed with one-way ANOVA. ** *p* < 0.01, *** *p* < 0.001, ^###^
*p* < 0.001 for significant difference between treatments in comparison to control medium (Con.).

**Table 1 marinedrugs-18-00061-t001:** Relative expression of nAChR subunits mRNA (fold) in breast cancer cell lines compared with normal cell line HS578BST.

Cell Lines	nAChR Subunits								
Name	α3	α4	α5	α7	α9	α10	β2	β3	β4
MDA-MB-453	++	+	++	+	++	*ns*	++	+	+++
BT-549	+	*ns*	++	++	+++	*ns*	*ns*	+	+
MDA-MB-361	*ns*	*ns*	++	+	++	++	*ns*	*ns*	+++
HS578T	++	++	+	+	++	++	*ns*	++	+++
Bcap-37	++	++	+	+	++	+++	++	+++	+++
HCC1395	++	++	++	++	++	+++	++	++	+++
BT-483	++	++	++	+	++	+++	++	++	++
SK-BR-3	+++	+	++	++	++	*ns*	*ns*	++	+++
BT-20	+++	++	++	++	++	++	+	++	++
ZR-75-30	++	+	++	++	++	*ns*	++	++	+++
HCC1806	+++	++	++	++	++	++	++	++	+++
MCF-7	*ns*	++	++	++	++	++	*ns*	*ns*	+++
AU565	+++	+	++	++	++	+++	*ns*	+++	+++
HCC1937	+++	*ns*	++	++	+	*ns*	*ns*	+++	+++
MDA-MB-231	+++	++	++	+	+++	++	+	+++	+++
MDA-MB-157	++	++	++	++	++	+++	++	++	++
HS578BST	-	-	-	-	-	*-*	-	-	-

Comparisons of differently expressed of nAChRs in normal control cells (HS578BST) and breast cancer cells. +, ++ and +++ represent upregulation of expression (*p* < 0.05, 0.01 and 0.001, respectively); ns represented no significant difference.

**Table 2 marinedrugs-18-00061-t002:** Relative expression of nAChR subunits protein (fold) in breast cancer cell lines compared with normal cell line HS578BST.

Cell Lines	nAChR Subunits (Protein Level)									
Name	α3	α4	α5	α6	α7	α9	α10	β2	β3	β4
MDA-MB-453	*ns*	*ns*	*ns*	*ns*	*ns*	*ns*	*ns*	*ns*	*ns*	*ns*
BT-549	*ns*	*ns*	*ns*	*ns*	*ns*	+++	*ns*	*ns*	*ns*	*ns*
MDA-MB-361	*ns*	+	+	*ns*	*ns*	+++	*ns*	*ns*	*ns*	*ns*
HS578T	*ns*	*ns*	*ns*	*ns*	*ns*	+++	*ns*	*ns*	+++	*ns*
Bcap-37	*ns*	*ns*	*ns*	*ns*	*ns*	+++	*ns*	*ns*	++	*ns*
HCC1395	*ns*	*ns*	*ns*	*ns*	*ns*	*ns*	*ns*	*ns*	*ns*	*ns*
BT-483	*ns*	*ns*	+++	++	+	+++	*ns*	*ns*	+++	*ns*
SK-BR-3	*ns*	+	+	*ns*	*ns*	+++	*ns*	*ns*	*ns*	+
BT-20	*ns*	++	+	+	++	+++	*ns*	*ns*	+++	*ns*
ZR-75-30	++	+	++	+	+	+++	+++	++	*ns*	++
HCC1806	+++	+++	+++	+++	*ns*	+++	++	*ns*	++	*ns*
MCF-7	*ns*	*ns*	*ns*	*ns*	*ns*	+++	++	*ns*	*ns*	*ns*
AU565	+++	+++	+++	+++	+++	+++	+++	++	*ns*	+++
HCC1937	+++	*ns*	*ns*	*ns*	*ns*	+++	*ns*	*ns*	++	*ns*
MDA-MB-231	++	++	*ns*	*ns*	++	+++	*ns*	*ns*	*ns*	++
MDA-MB-157	+++	+++	++	+++	+++	+++	+++	*ns*	+++	+++
BT474	+++	*ns*	++	*ns*	+	+++	*ns*	*ns*	++	+
HS578BST	−	−	−		−	−	−	−	−	−

Comparisons of differently expressed of nAChRs in normal control cells (HS578BST) and breast cancer cells. +, ++ and +++ represent upregulation of expression (*p* < 0.05, 0.01 and 0.001, respectively); ns represented no significant difference.

**Table 3 marinedrugs-18-00061-t003:** Dose-effect relationship of αO-GeXIVA on the breast cancer cells and normal cells (24 h).

Treatment	IC_50_ (uM)	Control	11.25 μM	22.5 μM	45 μM	90 μM	180 μM
Cells		Mean ± SD					
MDA-MB-157	35.18	100 ± 2.82 ^d^	72.19 ± 3.53 ^c^	68.14 ± 2.44 ^c^	54.83 ± 3.27 ^b^	51.33 ± 3.23 ^b^	45.37 ± 2.91 ^a^
MDA-MB-231	125.1	100 ± 5.61 ^e^	88.07 ± 1.69 ^d^	85.88 ± 3.41 ^d^	78.92 ± 1.41 ^c^	65.65 ± 2.63 ^b^	39.93 ± 1.87 ^a^
MDA-MB-361	77.03	100 ± 5.35 ^c^	78.58 ± 6.30 ^b^	75.48 ± 6.82 ^b^	70.67 ± 6.32 ^b^	64.57 ± 6.77 ^b^	33.34 ± 1.84 ^a^
MDA-MB-453	65.96	100 ± 4.58 ^d^	93.14 ± 1.51 ^c^	91.72 ± 0.58 ^c^	81.52 ± 3.30 ^b^	75.41 ± 4.02 ^b^	65.47 ± 9.63 ^a^
HCC1395	73.50	100 ± 0.80 ^f^	77.79 ± 0.46 ^e^	73.15 ± 0.59 ^d^	68.21 ± 1.32 ^c^	57.47 ± 1.75 ^b^	49.93 ± 4.03 ^a^
HCC1937	127.8	100 ± 1.89 ^d^	94.11 ± 1.71 ^d^	93.92 ± 1.43 ^d^	80.53 ± 7.43 ^c^	67.17 ± 1.53 ^b^	28.66 ± 0.48 ^a^
HCC1806	110.7	100 ± 2.74 ^f^	88.83 ± 0.97 ^e^	78.62 ± 1.04 ^d^	60.32 ± 0.91 ^c^	27.76 ± 1.18 ^b^	9.86 ± 0.46 ^a^
BT20	113.1	100 ± 9.86 ^c^	87.15 ± 5.51 ^c^	72.38 ± 4.73 ^b^	69.44 ± 7.89 ^b^	62.18 ± 4.16 ^b^	39.76 ± 9.75 ^a^
BT474	93.63	100 ± 0.47 ^e^	92.98 ± 0.63 ^d^	91.94 ± 1.51 ^d^	77.65 ± 0.62 ^c^	23.95 ± 0.48 ^b^	1.51 ± 0.33 ^a^
BT483	103.7	100 ± 4.78 ^c^	92.85 ± 1.58 ^c^	92.02 ± 0.94 ^c^	80.73 ± 3.44 ^b^	74.37 ± 4.19 ^b^	63.75 ± 7.59 ^a^
BT549	71.08	100 ± 4.22 ^d^	73.59 ± 6.23 ^c^	61.09 ± 2.07 ^b^	59.93 ± 3.01 ^b^	53.59 ± 6.37 ^b^	21.90 ± 2.98 ^a^
SK-BR-3	91.16	100 ± 3.75 ^d^	84.69 ± 3.09 ^d^	79.04 ± 3.27 ^c^	72.08 ± 10.11 ^c^	42.47 ± 3.36 ^b^	12.37 ± 0.22 ^a^
ZR-75-30	88.55	100 ± 7.48 ^e^	62.04 ± 1.25 ^d^	54.80 ± 1.54 ^c^	53.02 ± 1.95 ^c^	28.72 ± 4.29 ^b^	8.66 ± 2.59 ^a^
Hs578T	94.04	100 ± 4.15 ^e^	73.88 ± 5.52 ^d^	70.56 ± 4.27 ^d^	63.17 ± 5.46 ^c^	40.01 ± 1.21 ^b^	2.11 ± 0.80 ^a^
MCF-7	105.6	100 ± 1.04 ^d^	93.65 ± 1.04 ^c^	91.78 ± 0.43 ^c^	89.33 ± 1.25 ^c^	70.25 ± 5.69 ^b^	42.30 ± 0.38 ^a^
Au565	109.2	100 ± 2.95 ^d^	72.38 ± 6.32 ^c^	67.98 ± 3.35 ^c^	57.67 ± 2.14 ^b^	51.28 ± 8.26 ^b^	2.04 ± 0.72 ^a^
Bcap-37	78.92	100± 2.22 ^f^	89.54 ± 3.70 ^e^	80.12 ± 4.72 ^d^	64.51 ± 3.71 ^c^	47.27 ± 3.03 ^b^	1.42 ± 0.13 ^a^
MCF-12A	~246	100 ± 5.67 ^e^	90 ± 4.72 ^d^	79.49 ± 4.46 ^c^	76.34 ± 0.71^bc^	68.71 ± 3.53 ^b^	53.88 ± 6.01 ^a^
MCF-10A	~280	100 ± 3.21 ^d^	89.68 ± 3.75 ^c^	88.82 ± 2.65 ^c^	86.07 ± 2.70 ^c^	72.92 ± 4.60 ^b^	57.90 ± 5.09 ^a^
HS578BST	~285	100 ± 3.42 ^c^	99.23 ± 0.75 ^c^	98.26 ± 1.81 ^c^	97.22 ± 0.18 ^c^	92.15 ± 3.87 ^b^	72.30 ± 3.27 ^a^

The lowercase letters a, b, c, d, e and f represent the 95% confidence interval. One-way ANOVA was performed on the same row (same cells) and different columns (different concentrations) in the table. It shows that values with same superscript letters in the same line are of no significant difference (*p* > 0.05), those with different letters are of significant or extreme difference (*p* < 0.05).

**Table 4 marinedrugs-18-00061-t004:** Molecular classification of the breast cancer/ normal cells.

Cell Lines	Molecular Classification				Years	Source and Disease
Name	Subtypes	ER	PR	HER2		
MDA-MB-453	HER2			+	1976	Female, 48 years, Caucasian, pericardial effusion, metastatic carcinoma
BT-549	Basal-like	−	−	−	1978	Female, 72 years, Caucasian, breast, ductal carcinoma [59]
MDA-MB-361	Luminal B	+	−	+	1974	Female, 40 years, Caucasian, brain, adenocarcinoma [60,61]
HS578T	Basal-like	−	−	−	1977	Female, 74 years, Caucasian, breast, carcinoma [62]
Bcap-37	Luminal A	+	+	−	1981	Female, 48 years, Chinese, breast, medullary carcinoma [63]
HCC1395	Basal-like	−	−	−	1998	Female, 43 years, Caucasian, breast/duct, ductal carcinoma [34,64]
BT-483	Basal-like	−	−	−	1978	Female, 23 years, Caucasian, breast, ductal carcinoma [65,66]
SK-BR-3	HER2	−	−	+	1970	Female, 43 years, Caucasian, pleural effusion, adenocarcinoma [67]
BT-20	Basal-like	−	−	−	1958	Female, 74 years, Caucasian, breast, carcinoma [68]
ZR-75-30	Luminal A	+	+	−	1978	Female, 47 years, Black, ascites, ductal carcinoma [69]
HCC1806	Basal-like	−	−	−	1998	Female, 60 years, Black, breast, squamous carcinoma [34,64]
MCF-7	Luminal A	+	+	−	1973	Female, 69 years, Caucasian, pleural effusion, adenocarcinoma [59,60,70]
AU565	HER2	−	−	+	1970	Female, 43 years, Caucasian, malignant pleural effusion, malignant pleural effusion [67]
HCC1937	Basal-like	−	−	−	1998	Female, 23 years, Caucasian, breast/duct, ductal carcinoma [64,71]
MDA-MB-231	Basal-like	−	−	−	1974	Female, 51 years, Caucasian, pleural effusion, adenocarcinoma [60,61]
MDA-MB-157	Basal-like	−	−	−	1972	Female, 44 years, Black, breast/medulla, medullary carcinoma [72]
BT-474	Luminal B	+	+	+	1978	Female, 60 years, Caucasian, breast/duct, ductal carcinoma [59,65,66]
MCF-12A	Normal				1992	Female, 60 years, Caucasian, breast, fibrocystic, disease [66]
MCA-10A	Normal				1984	Female, 36 years, Caucasian, breast, fibrocystic, disease [37,66]
HS578BST	Normal				1977	Female, 74 years, Caucasian, breast, normal [62]

ER: Estrogen receptor; PR: Progesterone Receptor; HER2: Human Epidermal Growth Factor Receptor2. MCF-12A, MCF-10A and HS578BST were considered human normal mammary gland epithelial cell lines, others were breast cancer cells.

**Table 5 marinedrugs-18-00061-t005:** List of the primers and their sequences.

Primer Name (nAChR Subunits)	Sequence	Product Size
α3-F	AACGTGTCTGACCCAGTCATCAT	139 bp
α3-R	AGGGGTTCCATTTCAGCTTGTAG	
α4-F	CGGCAGCCGCCGGATGAC	201 bp
α4-R	ATGAAATTCGGCTCCTGGACCTA	
α5-F	ACTCCACCGGCAAACTACAA	267 bp
α5-R	CAGGCGCTTGATTACAAATGA	
α6-F	GGCCTCTGGACAAGACAA	414 bp
α6-R	AAGATTTTCCTGTGTTCCC	
α7-F	CCACCAACATTTGGCTGCAA	218 bp
α7-R	TATGCCTGGAGGCAGGTACT	
α9-F	TGGCACGATGCCTATCTCAC	171 bp
α9-R	TGATCAGCCCATCATACCGC	
α10-F	TGACCTCTTTGCCAACTACAC	138 bp
α10-R	CACAGATACAGGGTCAGCAC	
β2-F	GGCATGTACGAGGTGTCCTT	152 bp
β2-R	ACCAAGTCGATCTCTGTGCG	
β3-F	GGTCCGCCCTGTATTACATTC	146 bp
β3-R	AGCGTCTAACTTGTGGTCTG	
β4-F	TCACAGCTCATCTCCATCAAGCT	100 bp
β4-R	CCTGTTTCAGCCAGACATTGGT	
GAPDH-F	CAGCCTCAAGATCATCAGCA	103 bp
GAPDH-R	TGTGGTCATGAGTCCTTCCA	

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
