# Peer review of "Differential Expression of Nicotine Acetylcholine Receptors Associates with Human Breast Cancer and Mediates Antitumor Activity of αO-Conotoxin GeXIVA"

_marinedrugs, 2020, doi:10.3390/md18010061_

Round 1

Reviewer 1 Report

The manuscript can be accepted.

Author Response

Thanks for reviewer’s positive comment.

Reviewer 2 Report

The manuscript has been greatly improved with respect to previous versions, and there are just a few items that need revision and could be incorporated in the final revised version of this manuscript.

In Line 18 (abstract) the term “analyzation” is an unusual term, should be checked ? Alternatively use the following sentence, “as determined by fllow cytometry and Western blotting, except for MDA-MB-453 and HCC1395 cell lines”

Line 33, It should be " and non-(alpha)-subunits

Line 41, it should be a-neurotoxins “reduced the"

Line 42, “Bychkov et al. using curare also confirmed that Ly6/uPAR proteins”

Line 129, delete “which”

Line 148, should be " levels than normal cell line "

Line 183 it should be "compared to"

Line 223,  it should be "The specificity"

Lines 367-369, It should be "So, this is the first time that it is examined each nAChR subunit gene expression mRNA) in so many (up to 17) breast cancer cell lines and compared " "

Check references and homogenize them. In some references titles of articles exhibit capital letters in others just the first letter of the title is in capital letters. Note also that ref. 55 lacks the greek letters.

Author Response

This manuscript is a resubmission of an earlier submission. The following is a list of the peer review reports and author responses from that submission.

Round 1

Reviewer 1 Report

General comments

This manuscript needs important English language editing and careful revision, before being acceptable for publication.

The introduction needs to focus on the main aims of the article, and should be shortened. The Material and Methods are sound and the various techniques used are reliable. However, statistical procedures used are poorly described (e.g. ANOVA). The results section also should be more concise, and a mechanistic result should be provided on the manner αO-conotoxin GeXIVA by blocking overexpressed α9α10 nAChRs inhibit proliferation of breast cancer cells and/or supportive pathway used. In the absence of such data the manuscript will be mainly descriptive and its impact on the field will remain low. In the conclusions, it is indicated that αO-conotoxin GeXIVA promotes apoptosis in vitro, however the cell markers used are not described?

Specific Comments

1 Lines 14-15, Indicate in the Abstract that HS578BST is a cell line, like this "with the normal epithelial cell line HS578BST”

2 Lines 15-16 are not clear. May be what authors want to say is the following: α9 nAChR was highly expressed in almost all the breast cancer cell lines in comparison to normal cells. The different expression is prominent (P < 0.001) with the exception of MDA-MB-453 and HCC1395 cell lines, as analized ...

Line 24, it should be: “Cancer is one of the three principal diseases causing human mortality, and was responsible for 8.8 million deaths in 2015.” Line 25, The 8.8 million deaths of cancers is just for the USA? Line 27, "after lung cancer, and also..... (add and ). Line 29, 410 thousands women died of breast cancer every year all over the world....... (delete "were"; delete "in") Line 37, It should be: “nAChRs are much more complicated, and composed of different α and β subunits to form various” Line 40, Note that reference 10 is the same as reference 6. Therefore delete one of the references in the list of reference and re-number the list. line 46, Note that reference 14 has no volume or page number in the list of references. Line 50, define SCLC Lines 51-53 the content is not clear, Line 57, "on the pharmacology" has no sense. Rephrase the sentence starting in Line 56 Line 58-59, it should be "which is distributed " Lines 59-60, the jargon used is just for specialist in the field, and should be revised. For non specialist is not clear what is meant by ND96. In addition is not clear in which tissue data was obtained. I know that was obtained in oocytes but this is not indicated. In addition, since IC50 values are very similar for the extracellular medium that contained Ba2+ or Ca2+, I suggest just to indicate "with IC50 values in the low nanomolar range [26]"

Note that name of authors is not given in the [26] quoted reference. Please give full reference in the list.

Line 61, it should be "in a rat model" Line 62 replace shown by "showed". Line 66, It should be: "and the nAChR subunits that may be potential targets for breast"….. Line 75, Note that in the cladogram of Figure 1, the epsilon nAChR subunit was not represented.

Also, note in the figure legend of Fig. 1, it is indicated “There are 15 different human nAChR subunits”, please count the subunits you have shown in the cladogram!!

Line 136, It should be "Moreover, previous studies showed that nAChRs play a crucial" Line 137, Note that quoted reference 32 lacks volume and page numbering of the journal (list of references, complete) Line 155, it should be “which were statistically significantly different from the normal control “. Line 184, It should be "breast cancer cell lines expressing 2~20 fold higher levels of α3 nAChR subunit, including Line 212, It should be: "nAChR are highly expressed in all the tested human breast cancer cell lines "compared" to normal Line 253, it should be: "For MDA-MB-157 cells, all investigated functional nAChRs were "expressed" on "the" cell surface, but the Lines 259-261, It should be "It is worth noting that the α9 nAChR "was" highly expressed in almost all the breast cancer cell lines, "when compared to normal  cells", "differences were prominent (p < 0.001) except for the MDA-MB-453 and HCC1395 cell lines (Table 2)." In Figure 4A, label the ordinate axis Line 281, it should be "Western blot"; What is (Santa)? Line 282, It should be: "protein content from each cell line was assessed. (You have already said that you are going to use Western blotting). Line 289, it should be “BT483, MDA-MB-231 and MDA-MB-157 cell lines that showed significant” Line 308, it should be "Western blot" Line 316, it should be "Figure 5 and Figure 6), and we reported that αO-conotoxin GeXIVA[1,2] was a highly potent and selective Line 323, it should be "breast cancer cell lines," Line 362-363, It should be "which contain nearly all 363 types of breast cancer cells known so far. Line 386-387, it should be “In this study, the breast epithelial cell line HS578BTS was used as control.” Line 402, “In some cases, mRNA degrades rapidly even if it is transcribed,” Line 475, it should be "Dulbecco's"

Explain what is the meaning of α*-conotoxin

Reference 6, lacks title of the journal, but is repeated as reference 10, please amend.

38 Reference 14, add volume and page numbers

Reference 24 and 25 correct title of the articles Reference 26, add authors names, and page numbers Reference 30, add author names Reference 32, add volume and page numbers Reference 56, add author names Reference 66, add author names

Reviewer 2 Report

The work of Sun Z et al is devoted to the important and interesting theme: the role of nicotinic receptors in the epithelial cancer development and searching of new compounds for the cancer treatment.

In spite the high relevance of the work and the great volume of obtained results, there are several serious remarks.

Recently, the work devoted to the expression of different nAChR subunits and endogenous peptide Lynx1 in the different cancer cell lines was published (Bychkov et al., PLOS One, 2019).

In that work, expression of a3 and b2 subunits on the mRNA level in MCF-7 line was detected. Moreover, expression of functional a7-nAChRs in MCF-7 cells was demonstrated by confocal microscopy. How the authors could explain contradiction between their data and that manuscript?

The irreversible binding of alphaO-conotoxin to the nAChRs should be discussed. What about lethal doses of this toxin? Irreversible binding usually is accompanied with high systemic toxicity, what do the authors think about it? Breast cancers are often Her2+, please discuss possible relationships between overexpression of this receptor and nAChRs, especially in line with the recent study pointing on possible complex formation between a7-nAChR and other receptors from the HER family (Chernyavsky AI et al, 2015) The data from the Table 2 must be supported by the data of Western-blot analysis

Minor points:

The figure 1 is not related to the theme of the manuscript. I believe it should be removed The data from the table 3 will be more informative in the form of dose-response curves. The specificity of antibodies used should be confirmed The references 14 and 32 are identical

Round 2

Reviewer 1 Report

General

The manuscript has been improved despite that NO answer to the previous comments were given:

(i)the Introduction has not been shortened and focused, as suggested,   (ii) The results section has not been made more concise, as suggested,   (iii) No mechanistic or supportive pathway is given on the manner αO-conotoxin GeXIVA by blocking overexpressed α9α10 nAChRs inhibits proliferation of breast cancer cells. This lack of information certainly will reduce the impact that this manuscript will have on the field.   (iv) In addition, there are still some editing problems and consistency in the edition, as shown here below.,   (vI) It would be fair that authors may cite some review articles related to alpha-conotoxins that have been published in Mar Drug related to alpha-conotoxins, since this will keep the impact factor of the journal.

Minor points

Figure x (a space should be present between Figure and the number. Correct in all text. A space should be present between p the symbol < and the number 0.001 Also a space should be present in n = 3 Line 120 it should be "One microgram" line 286 it should be "that showed significance" or “that were statistically significant p < 0.05” Line 313, delete nicotinic acetylcholine receptor, and use abbreviation nAChR Line 324, suppress space between αO- and GeXIVA line 347, it should be "in comparison to control medium (Con) Line 387 delete point after Figure and add space before 2 Also in line 388 (S1, Figure.S1), also delete the repeated Figure S1 Line 397, WB has been defined? Line 403, Ca superscript for 2+, idem for line 409, Idem line 410 Line 414, instead of “to be higher » use “to be more” expressed Iine 416, secondhand ? Line 453, add space between less and than Line 475, it should be "receptor"2 Line 506, it should be "Western" Idem line 513, "Western" There are many other inconsistencies in the edition that merit to be revised by authors or editors.

Reviewer 2 Report

Response to Reviewer 2 Comments

Point 1: Recently, the work devoted to the expression of different nAChR subunits and endogenous peptide Lynx1 in the different cancer cell lines was published (Bychkov et al., PLOS One, 2019). In that work, expression of a3 and b2 subunits on the mRNA level in MCF-7 line was detected. Moreover, expression of functional a7-nAChRs in MCF-7 cells was demonstrated by confocal microscopy. How the authors could explain contradiction between their data and that manuscript?

Response 1: This question is reasonable, and we are sorry for the ambiguity of our statements. We should first of all recognize that the α3, α4, α7, α9 and β2 nAChRs gene are expressed in human breast adenocarcinoma MCF-7 cells, and our study got identical results. However, in this study we found that MCF-7 cells do not show significant changes in α3 and β2 nAChR expression compared to normal epithelial cells HS578BST. Similarly, Compared with normal cell line HS578BST, MCF-7 cells did not show significant changes in α7 nAChR expression. So, in our work, it's not that the α3, a7 and β2 nAChRs no expression, but there was no significant difference in expression of nAChRs between cancer cells and normal cells.

Reviewer:

It should be reflected in the Title of the Table in clearer manner that there are data relative expression in HS578BST line. Please, add the short discussion of other works (including Bychkov et al., 2019) devoted to the analysis of nAChR expression in cancer cells.

Point 2: The irreversible binding of alpha O-conotoxin to the nAChRs should be discussed. What about lethal doses of this toxin? Irreversible binding usually is accompanied with high systemic toxicity, what do the authors think about it? Breast cancers are often Her2+, please discuss possible relationships between overexpression of this receptor and nAChRs, especially in line with the recent study pointing on possible complex formation between a7-nAChR and other receptors from the HER family (Chernyavsky AI et al, 2015) The data from the Table 2 must be supported by the data of Western-blot analysis

Response 1: Thank you for pointing this out. Indeed, there were debates which permeate the study of nAChRs and antagonists. α-Bungarotoxin is a specific antagonist of α7 nAChR. α Bungarotoxin binds irreversibly to the nAChR and thecomplex bungarotoxin-nAChR is immunogenic, which may result in production of autoimmune antibodies. For this reason, α bungarotoxin is a suitable experimental model for myasthenia gravis[1]. Moreover, conotoxins like the ω-Conotoxins GVIA, often irreversibly to the N-type voltage-gated calcium channel α1 subunit[2]. However, α connotoxins from cone snail Conus consors abbreviated as CnIA with sequency GRCCCHPACGKYYSC and amidated C terminus are  selective and reversible antagonists of α7 nAChR[3]. For αO-GeXIVA, also is a selective and reversible antagonists of α9α10 nAChR, Luo SL et al. confirmed that blockade of α9α10 nAChRs by GeXIVA is reversed after 2 min of toxin washout by a two-electrode voltage clamp[4]. In our previous researches, GeXIVA displayed potent alleviation of neuropathic pain in a rat model[5]. For cytotoxicity, our preliminary results also showed that GeXIVA is less toxic to normal cells than to cancer cells. 50% of breast cancer cells were inhibited when the concentration of GeXIVA was close to 30~120 μM, however, a higher concentration (~300 uM) of GeXIVA could kill the same number of normal breast epithelial cells.

HER2 is a member of the epidermal growth factor receptor (EGFR) family. Other family members include EGFR or HER1, HER3 and HER4. HER-2 tyrosine kinase receptor is overexpressed in approximately 25% of invasive breast cancers[6,7]. Chernyavsky AI et al. confirmed that the growth-promoting effect of nicotine mediated by activation of α7 cm-nAChR synergizes mainly with that of epidermal GF (EGF), α3-vascular endothelial GF (VEGF), α4-insulin-like GFI (IGF-I) and VEGF and α9-EGF, IGF-I and VEGF in lung cancer cells[8]. In my study, we found that a9-nAChR was preferentially overexpressed in human breast tumor tissue in comparison with normal cells. Maybe, the growth-promoting effect of HER-2 and nAChRs is occurred in breast cancer. But, we have no way to prove it now.

The data from the Table 2 were acquired by using flow cytometry combined with fluorescent antibody labeling. The Material and Methods are sound and the various techniques used are reliable. In addition, we probably won't do the work in a short period.

Reviewer:

It is very serious point to support the data of flow-cytometry by Western-blot analysis. It is well known, that antibodies against different nACHR subunits are usually very non-specific, and it is a great trouble for investigators. It is slippery way to make the conclusions about relative protein expression only using antibodies with unknown specificity. That why the data should be supplemented by (1) Western-blot analysis, and (2) the confocal images pointing on specificity of used antibodies (the lines with and without nAChRs expression should be used).

Minor points: The figure 1 is not related to the theme of the manuscript. I believe it should be removed. The data from the table 3 will be more informative in the form of dose-response curves. The specificity of antibodies used should be confirmed. The references 14 and 32 are identical.

Response for minor points: The figure 1 not only described the classification of human nAChR subunits, but also reflected the true evolutionary relationship of nAChR subunits genes from homo species. For the data from the table3, dose-response curves maybe will be more informative, but too much of curves in the graph cause look like very disorder. The specificity of antibodies used has been confirmed (Page15, Lines 548-557). The references 14 and 32 have modified to references 13.

Reviewer:

Sorry, I did not find mentioned confirmation (Page15, Lines 548-557). Please, check the manuscript.

Round 3

Reviewer 2 Report

Point 2: It is very serious point to support the data of flow-cytometry by Western-blot analysis. It is well known, that antibodies against different nACHR subunits are usually very non-specific, and it is a great trouble for investigators. It is slippery way to make the conclusions about relative protein expression only using antibodies with unknown specificity. That why the data should be supplemented by (1) Western-blot analysis, and (2) the confocal images pointing on specificity of used antibodies (the lines with and without nAChRs expression should be used).

Response 2: This question is reasonable, and thank you for these precious comments and suggestions. We analysed the different expression of nAChRs by flow-cytometry, and the material and methods are sound and the various techniques used are reliable. Next, in the study, the antibodies against different nAChR subunits are specific (The specificity of antibodies used has been confirmed, in lines 537~546). As you say, maybe Western-blot analysis and confocal images are even more persuasive. The role and function of nAChRs in inhibiting the breast cancer cells are under investigation in our laboratory. Unfortunately, results are unavailable at this point.

Reviewer:

 Sorry, but I did not find yet the requested information in the manuscript. The lines 537-546 are devoted to the description of statistical analysis and conclusions. There were no any words about the specificity of antibodies used.

Ok, if the authors can not present now the data on the WB analysis, they can study the specificity of antibodies used on the cells without nAChR expression, for example HEK293T. It is a quite simple experiment and will not take much time. Without the data of this experiment confirmed by flow cytometry or confocal microscopy I can not accept this manuscript.  

Point 3: Sorry, I did not find mentioned confirmation (Page15, Lines 548-557). Please, check the manuscript.

Response 3: We are sorry that the some mistakes in the paper due to our careless and negligence. The specificity of antibodies used has been re-confirmed. Lines 537~546.

Reviewer: No information about antibodies specificity is available on the Page 15 yet.
